# Posterior Shoulder Dislocation with Engaging Reverse Hill–Sachs Lesion: A Retrospective Study of Ten Patients Treated with Arthroscopy or Open Reduction and Stabilization

**DOI:** 10.3390/jcm10071410

**Published:** 2021-04-01

**Authors:** Giorgio Ippolito, Michele Zitiello, Giancarlo De Marinis, Fabio D’Angelo, Michele F. Surace, Mario Ronga, Vincenzo Sepe, Luca Garro, Luca Faoro, Sergio Ferraro

**Affiliations:** 1Dipartimento di Scienze e Biotecnologie Medico Chirurgiche (DSBMC), Sapienza Università di Roma, Polo Pontino ICOT, 04100 Latina, Italy; 2Istituto Chirurgico Ortopedico Traumatologico (ICOT), 04100 Latina, Italy; dr.michelezitiello@gmail.com (M.Z.); ortodemarinis@gmail.com (G.D.M.); v.sepe@giomi.com (V.S.); 3Ospedale di Circolo, Fondazione Macchi “Università Insubria”, 21100 Varese, Italy; fabio.dangelo@uninsubria.it (F.D.); faoro.l@gmail.com (L.F.); sergio.ferraro.ortop@asst-settelaghi.it (S.F.); 4Dipartimento di Biotecnologie e Scienze Della Vita, Università Insubria, 21100 Varese, Italy; michele.surace@uninsubria.it; 5Azienda Ospedaliera Policlinico Universitario “G. Martino”, 98124 Messina, Italy; mronga@unime.it; 6Casa di Cura Villa Betania Giomi SpA, 00165 Roma, Italy; garro.luca@gmail.com

**Keywords:** shoulder, posterior dislocation, reverse Hill–Sachs lesion, McLaughlin surgery

## Abstract

This study compares two surgical techniques used to treat patients with posterior shoulder dislocation with an engaging reverse Hill–Sachs lesion. We assessed ten patients who were treated at the Surgical Orthopedic and Traumatological Institute (ICOT) of Latina and the Clinic of Orthopedic and Traumatological Surgery of the ASST Sette Laghi of Varese between 2016 and 2019. The patients were divided into two groups: the first comprising six patients who underwent the open surgery McLaughlin procedure as modified by Neer, the second including four patients who underwent the arthroscopic McLaughlin procedure. All patients received postoperative rehabilitation to achieve the best possible functional recovery of the affected shoulder. We then assessed the shoulder range of motion, the pain level, and the impact on quality of life with four tests: the Constant Scale, the Simple Shoulder Test (SST), the OXFORD Scale, and The University of California—Los Angeles (UCLA) Shoulder Scale. The mean scores of the first group were: 81.3 ± 9.8 SD (Constant Scale), 10.8 ± 1.06 SD (SST), 42.5 ± 5.4 SD (Oxford Scale), 30.8 ± 3.02 SD (UCLA Shoulder Scale); we calculated the following mean scores in the second group: 80.25 ± 4.1 SD (Constant Scale), 11.5 ± 0.8 SD (SST), 42 ± 4.06 SD (Oxford Scale), 32 ± 2.9 SD (UCLA Shoulder Scale). We found no significant differences between the two groups.

## 1. Introduction

The posterior dislocation of the glenohumeral joint is a rare pathology accounting for less than 5% of all shoulder dislocations. In approximately half of the cases, the pathology is due to a single trauma caused by a direct force exerted on the shoulder in the anteroposterior direction or by indirect forces associated with positions of internal rotation, adduction, and flexion of the shoulder; subluxations are other causes that are common in certain activities such as contact or overhead sports. It may also result from violent muscle contractions due to, for example, seizures or electrocution [1]. Atraumatic cases have been associated with genetic disorders of the connective tissues, such as the Marfan Syndrome, or with skeletal abnormalities, such as glenoid hypoplasia and retroversion [2].

Patients with posterior dislocation of the shoulder may have additional injuries that complicate their assessment and treatment. Among them, the reverse Hill–Sachs lesion, also called the McLaughlin lesion, is present in up to 50% of cases. The reverse Hill–Sachs lesion is the fracture of the anteromedial portion of the humeral head as a result of its posterior dislocation on the glena. Another injury that may occur is the Kim’s lesion, i.e., the detachment of the posteroinferior labrum with avulsion of the posterior capsular periosteum. Sometimes the trauma can be so severe that it involves the glenoid bone [3].

We assessed ten patients with posterior dislocation of the glenohumeral joint who were treated at the Surgical Orthopedic and Traumatological Institute (ICOT) of Latina and the Clinic of Orthopedic and Traumatological Surgery of the ASST Sette Laghi of Varese between 2016 and 2019. We divided the patients into two groups: the six patients of the first group underwent open surgery following the Neer’s method, while the four patients of the second group underwent the arthroscopic McLaughlin procedure. Even though our patient sample was small due to the rarity of the lesion, it gave us important insights into a pathology that can be easily overlooked even by the most attentive clinicians and that, if not treated on time, can affect the joint function and the vascularity of the humeral head; in addition, it can lead to chronic instability, osteonecrosis, and osteoarthritis [4].

## 2. Materials and Methods

We recruited ten patients who underwent surgery at the Surgical Orthopedic and Traumatological Institute (ICOT) of Latina and the Clinic of Orthopedic and Traumatological Surgery of the ASST Sette Laghi of Varese between 2016 and 2019. At the time of admission to the hospital, the patients were diagnosed with irreducible posterior dislocation of the shoulder with associated reverse Hill–Sachs lesion. Four patients had previously gone to other hospitals, where they did not get a diagnosis as the injury was investigated only with an anteroposterior (AP) view [5]. All our patients underwent computed tomography scan (CT scan) and attended a follow-up 12 weeks after surgery.

We chose the surgical technique based on the timing of the trauma, the percentage of bone loss, and the imaging depth. We divided the patients into two groups: the first group consisted of the six patients that underwent the Neer modification of the McLaughlin procedure, an open surgery; the second group was composed of the four patients that underwent the arthroscopic McLaughlin procedure.

The McLaughlin procedure is used for posterior dislocations with moderate humeral head defects. The surgery involves the tenodesis of the subscapularis tendon into the defective area, and it is performed arthroscopically with the patient in the semi-sitting or lateral decubitus position. After opening the arthroscopic access, the hematoma is removed and the long head of the biceps tenotomy is performed. The tendon of the subscapularis muscle is then detached from the lesser tuberosity [6]. Debridement of the anterior bone defect of the humeral head is carried out, a 5.5 mm metal suture anchor is applied (Figure 1), and the sutures are passed through the subscapularis tendon, relaxed, and placed with a sliding knot in the bony defect (Figure 2). The suture is reinforced by pushing OUT/IN with another suture placed lateral to the greater tubercle (Figure 3).

The Neer modification of the McLaughlin procedure (Figure 4) involves first the osteotomy of the lesser tuberosity: the biceps tendon is used to identify the bicipital groove, then the long head of the biceps tenotomy is performed and the lower edge of the subscapularis tendon is used to mark the lesser tuberosity; the osteotomy of the lesser tuberosity is performed from lateral to medial, starting from the biceps groove. Then, the bone fragment along with the subscapularis tendon is transplanted into the reverse Hill–Sachs lesion to reduce the dislocation; one or two 4 mm cannulated screws are then used to fix the lesser tuberosity and the attached subscapular tendon (Figure 5). The Neer’s modification provides additional support for the area of bone deficit and is preferred for all those patients whose lesion is more than three weeks old and when the reduction appears irreducible and dangerous.

In the postoperative period, patients of both groups wore an external-rotation shoulder brace for 40 days. In the months following the surgery, the patients were followed up at the Surgical Orthopedic and Traumatological Institute (ICOT) of Latina and the Clinic of Orthopedic and Traumatological Surgery of the ASST Sette Laghi of Varese, where they underwent post-surgery rehabilitation programs to achieve the best possible functional recovery of the affected shoulder.

We then assessed the shoulder range of motion, the degree of pain and disability, and the impact on patients’ quality of life with the most widely used orthopedic tests: the Constant Scale, the Simple Shoulder Test (SST), the OXFORD Scale, and The University of California—Los Angeles (UCLA) Shoulder Scale [2]. These tests are well-validated and provide a complete picture of the health status and functionality of the injured shoulder over time. The patients attended a medical checkup and filled out the questionnaires of the four tests; they also allowed us to keep track of their recovery and photographic records. The authors state that consent was obtained from all patients studied. The data were collected in a table and analyzed with the statistical software “XLSTAT”.

## 3. Results

In this mid-term retrospective study, the patients filled out the questionnaires of four tests for orthopedic assessment (Constant Scale, SST, OXFORD Scale, UCLA Shoulder Scale); we then compared their scores, and especially the shoulder range of motion, with the standard values of the general population. The tests were used to assess the kinematics of the operated limb (by measuring anterior flexion, extension, abduction, and intra and external rotation of the shoulder), the level of pain during the day or at night, and the ability to carry out daily and work activities as well as simple household chores and personal hygiene activities. This allowed us to understand the well-being of the patients in the months following the surgery. The patients were between 37 and 64 years old at the time of injury. We analyzed the data and obtained the following mean scores from the group that underwent the Neer modification of the McLaughlin procedure: 81.3 ± 9.8 SD (Constant Scale), 10.8 ± 1.06 SD (SST), 42.5 ± 5.4 SD (Oxford Scale), 30.8 ± 3.02 SD (UCLA Shoulder Scale). The mean scores of the second group, which underwent the McLaughlin procedure, were as follows: 80.25 ± 4.1 SD (Constant Scale), 11.5 ± 0.8 SD (SST), 42 ± 4.06 SD (Oxford Scale), 32 ± 2.9 SD (UCLA Shoulder Scale).

We found no significant differences between the scores of the group that underwent open surgery and the scores of patients that underwent the arthroscopic procedure. Table 1, Table 2 and Table 3 and Figure 6 and Figure 7 summarize the examined criteria, the data, and the patients’ scores collected during the checkup. We compared the Constant shoulder scores in the literature with the scores of our patients, finding no significant differences in the values (Table 3).

## 4. Discussion

Posterior dislocation is considered rare as it accounts for less than 5% of shoulder dislocations. This type of lesion is not easy to diagnose since standard examinations, such as anteroposterior and axial radiographs, do not always identify it. Indeed, in 60% to 80% of the cases, posterior dislocations of the shoulder are not diagnosed at the first clinical examination as the patients’ pain does not allow joint mobilization and the axial view, which could be diagnostic, cannot be taken. Therefore, it is recommended to perform the Y-scapular view that provides information about the position of the humeral head and the severity of the reverse Hill–Sachs lesion: the humeral head can be clearly seen posterior to the glenoid and the Hill–Sachs lesion, which is typically seen in the anteromedial part of the humeral head, can also be studied [7].

The posterior dislocation of the shoulder often causes bone injuries, such as fractures of the anterior part of the humeral head (also known as reverse Hill–Sachs lesions), posterior labrum tears, and posterior glenoid rim fractures. The clinical treatment depends on a number of anatomical and functional factors that must be evaluated in each patient. The treatment can be either nonoperative or surgical: The nonoperative approach is adopted when the dislocation is stable after a reduction maneuver and when there are no significant bone deficits; after the maneuver, the shoulder is immobilized for a limited period of time [8,9]. On the other hand, surgical treatments are preferred when the trauma has caused joint instability or lesions of bone or soft tissues that require surgical stabilization. The choice of the operative technique depends on the joint defect: anatomical reconstruction of the articular surface is recommended for deficits between 25 and 40%, while prosthetic surgery is usually preferred for injuries that reduce the shoulder mobility by more than 40%. In 2017, the French Society of Arthroscopy [8] compared the clinical outcomes of nonoperative and surgical treatments in a prospective multi-center non-randomized study, and they showed that operative techniques have better outcomes.

In patients with Hill–Sachs lesion, a closed reduction under local or general anesthesia is usually chosen when the humeral bone loss is less than 25%. The reduction is achieved by applying longitudinal traction and direct posterior-anterior pressure on the humeral head. The arm is then rotated 15 degrees and immobilized in 15 degrees of extension for four to six weeks; this position relaxes the capsule posteriorly and accelerates the healing process [9]. However, this maneuver could lead to inveterate irreducible dislocations and surgical neck fractures. Moreover, the closed reduction could cause posterior shoulder instability and hinder the reduction of the humeral head, exacerbating its fracture. Open reduction is usually performed with an anterior deltopectoral approach, allowing the evaluation of the humeral head fracture that is disengaged from the posterior glenoid edge. When considering the type of injury and the treatment, the possibility of redisplacement, nerve injury, intraoperative fracture, and humeral head replacement during surgery should always be considered [10].

Fractures that compromise 20% to 40% of the joint surface of the humeral head could be treated with a closed reduction, but larger fractures have a higher chance of recurrence and should be treated with the McLaughlin procedure or the Neer’s modification that involve the transfer of the subscapularis tendon alone or together with the lesser tuberosity, respectively [11,12]. Prosthetic replacement, derotation osteotomy of the humeral neck, and humeral head reconstruction with allograft or autograft transplantations have also been proposed for osteochondral defects larger than 45%. In a recent systematic review, Basal O. et al. [13] have pointed out that the McLaughlin arthroscopic technique should be preferred in the case of acute irreducible posterior dislocations (less than three weeks after injury) with low bone loss (less than 20%), even if the open surgery as described by Neer gives similar functional outcomes. In 2020, Georgios Paparoidamis et al. [14] highlighted how the Neer’s modification leads to better functional results compared to arthroscopic treatment in cases of chronic posterior dislocations (more than three weeks after injury) with less than 40% bone loss.

We compared the Constant shoulder scores in the literature with the scores of our patients, finding no significant differences in the values. We can therefore conclude that the recovery of our patients, in terms of functional outcomes and pain level, is comparable to that of patients treated in other countries, confirming the validity of the techniques we use to treat posterior shoulder dislocations with associated Hill–Sachs lesions [15]. The major strength of the study is the number of patients: in fact, posterior shoulder dislocation is one of the rarest shoulder injuries and its diagnosis is often missed. The limitation is not having compared the results of the surgical technique with nonoperative treatment.

The bias is that the surgeries were performed by two different surgeons. The first surgeon (Latina) preferred the arthroscopic technique while the second (Varese) preferred the open technique.

## 5. Conclusions

Posterior shoulder dislocation is one of the rarest shoulder traumas and its diagnosis is often missed. In Latina and Varese, ten patients were diagnosed with posterior dislocation and required surgical treatment between 2016 and 2019. Following postoperative rehabilitation, all patients, which underwent either the McLaughlin procedure or the Neer’s modification depending on their injury, had good recovery of the shoulder range of motion. However, we noticed that the acute arthroscopic surgery resulted in a greater feeling of stability and well-being compared to the open surgery (Figure 6 and Figure 7), while we found no differences in the subscapularis strength recovery [16]. This demonstrates how the timing of injury, bone loss, and general conditions are crucial to choosing the treatment [17]. We recommend the arthroscopic McLaughlin procedure for patients with recent irreducible dislocations (less than three weeks after injury) and small bone deficiencies (less than 20%) [18]. On the other hand, we suggest the Neer’s modification for patients with chronic irreducible dislocations (more than three weeks after injury) and bone loss between 20% and 40% [19]. In conclusion, the authors say more clinical randomized controlled multicenter are needed to be able to say which surgical technique is superior.

## Figures and Tables

**Figure 1 jcm-10-01410-f001:**
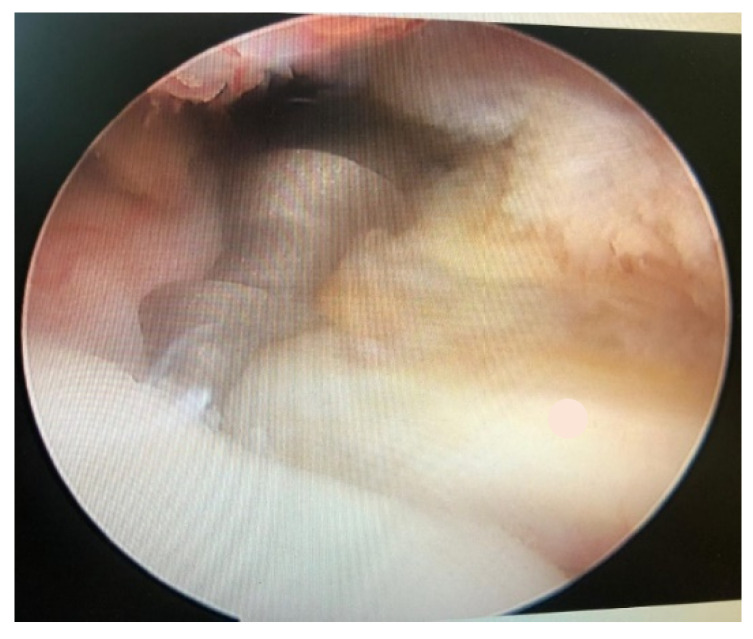
5.5 mm metal suture.

**Figure 2 jcm-10-01410-f002:**
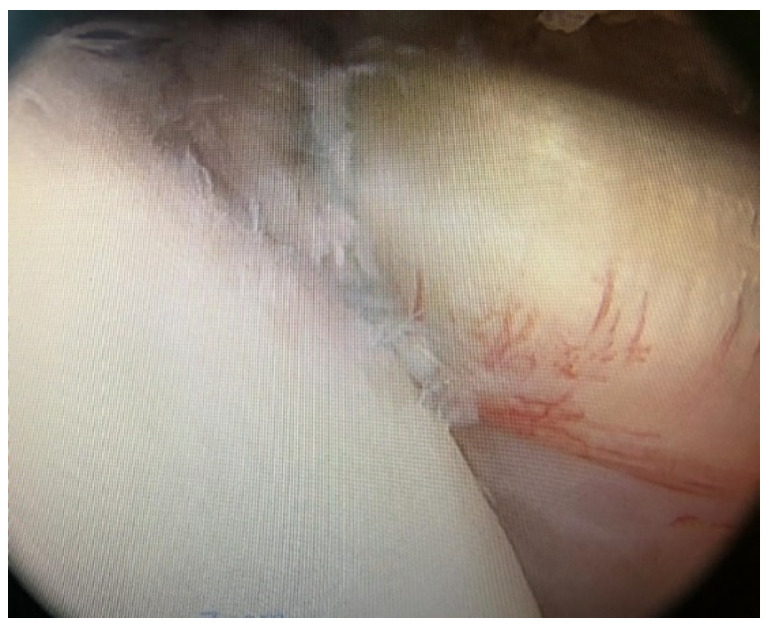
Tendon of the subscapularis muscle positioning in bone.

**Figure 3 jcm-10-01410-f003:**
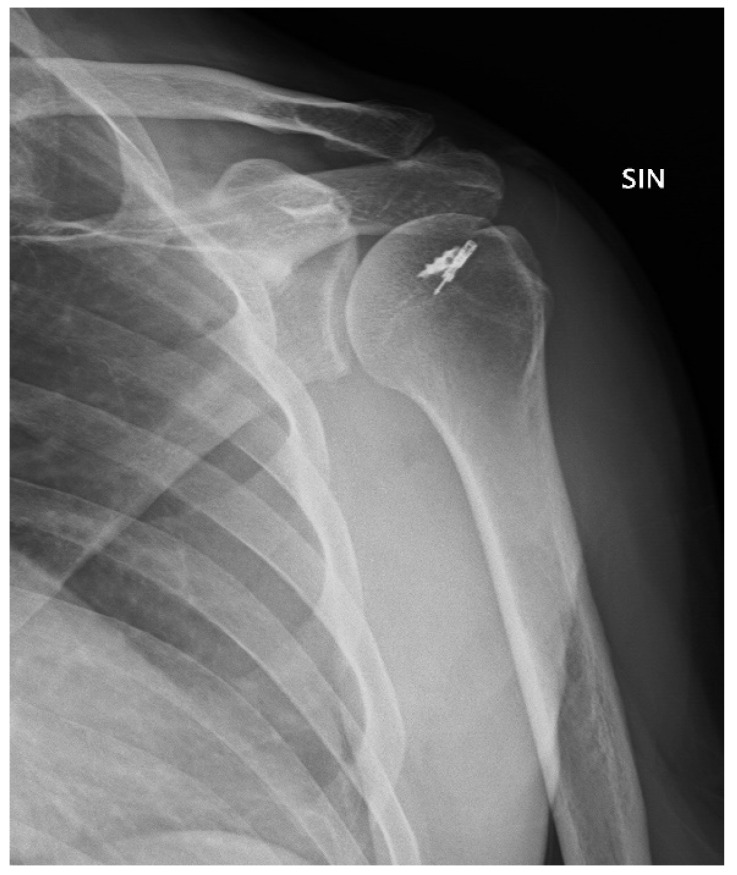
Post-operative X-ray.

**Figure 4 jcm-10-01410-f004:**
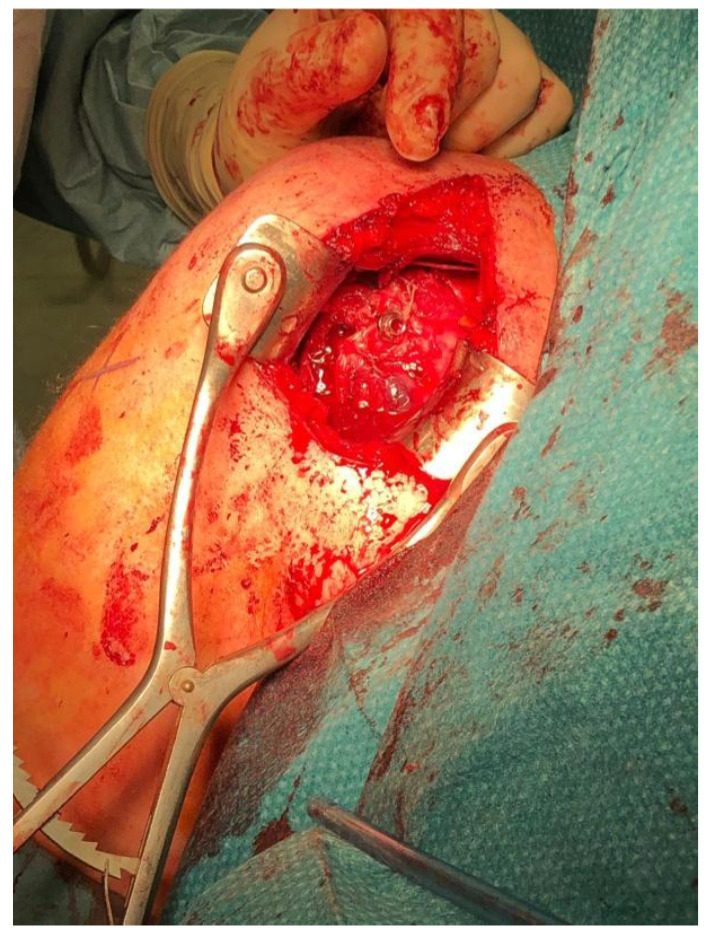
Open surgery Mclaughlin procedure as modified by Neer.

**Figure 5 jcm-10-01410-f005:**
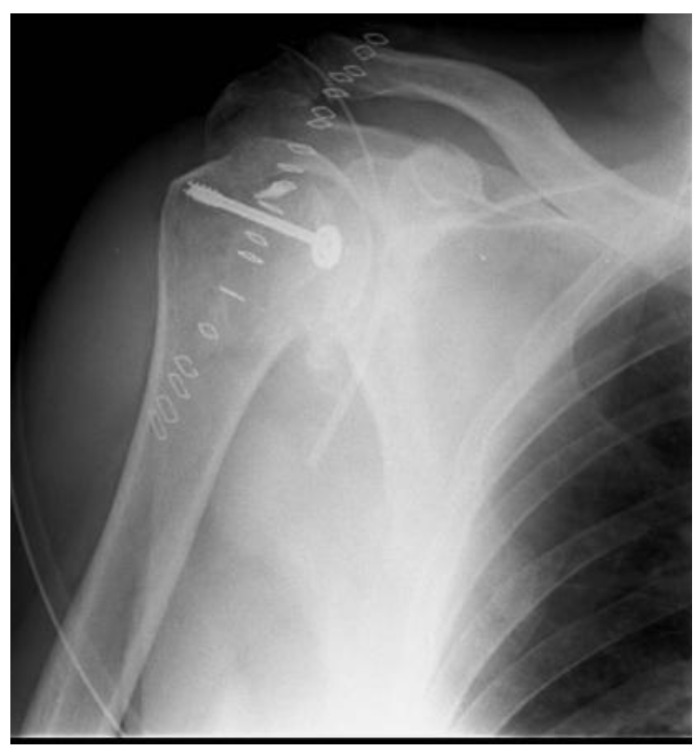
Post-operative X-ray.

**Figure 6 jcm-10-01410-f006:**
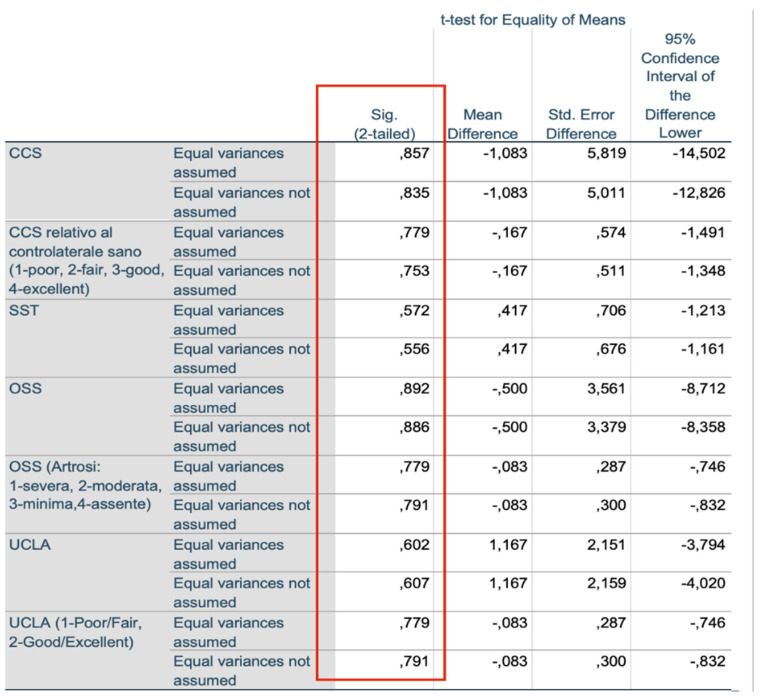
Independent sample test.

**Figure 7 jcm-10-01410-f007:**
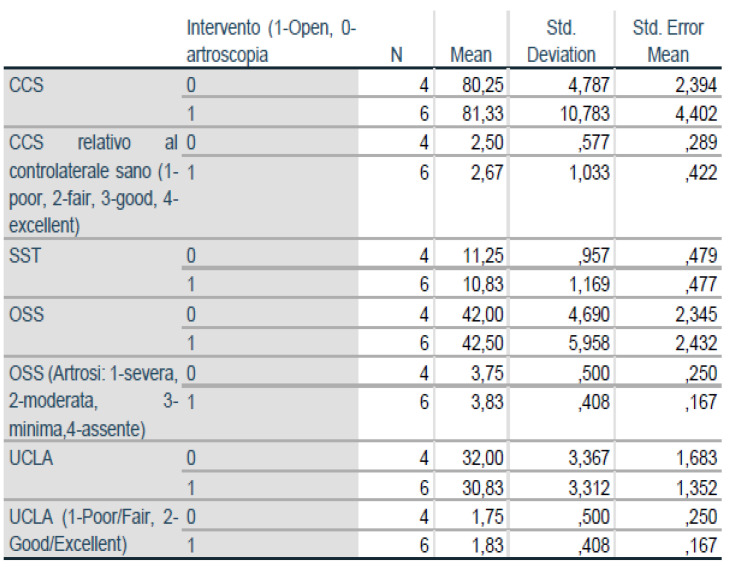
Group Statistics.

**Table 1 jcm-10-01410-t001:** Patients’ scores collected during the checkup.

Patient (First Group)	Constant Scale	SST (Simple Shoulder Test) Scale	Oxford Scale	UCLA (University of California—Los Angeles
F.G. (53)	63	11	32	25
C.M. (44)	75	9	41	30
F.P. (64)	92	12	48	34
D.M. (39)	82	10	41	31
B.D. (46)	89	12	46	34
F.P. (62)	87	11	47	31

**Table 2 jcm-10-01410-t002:** Patients’ scores collected during the checkup.

Patient (Second Group)	Constant Scale	SST (Simple Shoulder Test) Scale	Oxford Scale	UCLA (University of California—Los Angeles
I.M. (58)	75	10	35	27
C.U. (39)	82	12	45	34
R.M. (37)	86	11	44	33
R.S. (38)	78	12	44	34

**Table 3 jcm-10-01410-t003:** Constant shoulder scores in the literature and scores of our patients.

	Our Study (2021)	Cruz-Ferreira and AL. (2017)	Clavert and AL. (2017)	N. Aydin and AL. (2019)	O. Basal and AL. (2018)
Constant Scale (Average Value)	80.75	78	82	79.25	82.5

## Data Availability

Please contact author for data requests.

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
