# Peer review of "Posterior Shoulder Dislocation with Engaging Reverse Hill–Sachs Lesion: A Retrospective Study of Ten Patients Treated with Arthroscopy or Open Reduction and Stabilization"

_jcm, 2021, doi:10.3390/jcm10071410_

Round 1

Reviewer 1 Report

Thank you very much for allowing me to review your interesting manuscript.

Summary: Authors conducted a study comparing two surgical techniques (open Neer's vs arthroscopic McLaughlin) to treat posterior shoulder dislocations with an engaging reverse Hill-Sachs lesion, and found no significant differences in ROM, pain and quality of life tests. The indications for the types of procedures and groups were based on timing of injury and bone loss, showing equal results of the tests used. Authors further compared their Constant scores to literature and found comparable results.

Pros: A much needed study on a rare pathology. As authors stated, it is a rare pathology that can be overlooked or missed, with negative consequences.

Cons: Low n of 10 patients. Unclear methods and comparisons.

Major comments:

Figures 1 and 2 have artifacts of Moiré patterns (black lines from camera's sensor). A higher quality image may improve the manuscript if available.

Would be worth explaining the statistical analysis further. What tests were used, What groups were compared (open vs arthroscopic, constant scores of both vs general population from literature, etc.) and significance set at what alpha.

First paragraph of results could be moved to Methods section.

The tables should show p-values from the statistical tests used, and what groups were being compared. Possibly creating a new table to compare Constant scores from your patient population and the literature scores you compared them to. Additionally, you state differences in conclusion (see below comment), would be good to show the data proving this.

In the conclusion you state you found differences in arthroscopic vs open surgery. I did not see this in the results. What metric are you using? And does it have a significant p-value to state a difference between the procedures?

Minor comments:

page 2; line 76: Define CLBO

page 2; line 80: "figura" misspelling.

Could use more literature on posterior shoulder dislocation treatment options. Also worth clearing up what is anterior vs posterior pathology in discussion.

Author Response

I have inserted further statistical analysis. I compared our Constant scores to the literature and found the results to be comparable. I have created new tables.

Reviewer 2 Report

M and M - Was consent obtained from participants in the study? If not it may not be publishable.

                - Line 60-61 - move this information to the Results section about the age of the patients.

Discussion - Lines 183 (twice) and Line 192 - the word "conservative" is used and the proper term is "nonoperative". Please rewrite.

                  - Line 194 - Do you mean REVERSE Hill-Sachs lesion?

                  - At end of the discussion there is no strengths and limitations section to this paper. Here the authors should discuss bias in the study examples being the assignments to patient groups - OPEN or ARTHROSCOPIC. Other factors to mention are the small size and retrospective nature of the study. Here as well, the authors need to mention that there needs to be a good clinical randomized controlled multicenter trial to answer this question.

Conclusion - Lines 228-235 - repeat this info as it is not helpful and not needed in the conclusion. 

                   - also the conclusion should mention that this area needs further study like a RCT.

Author Response

The consent of the participants has been obtained.   Line 194 - Do you mean REVERSE Hill-Sachs lesion? YES.  ]. In conclusion, the authors say more clinical randomized controlled multicenter are needed to be able to say which surgical technique is superior

Round 2

Reviewer 1 Report

The authors have appropriately addressed my comments. Thank you.

Author Response

THANK YOU FOR REVIEWING OUR ARTICLE.
BEST REGARDS

Reviewer 2 Report

The authors have made some of the changes suggested. However, many of the chnges were not done or ignored. 

1) Age of the patients (a sentence) should be placed into the Results section and taken out of he Methods (as recommended last review).

2) Line 229 - you mean REVERSE Hill-Sachs (as recommended last review).

3) There is no strengths and limitations section (as recommended last review). This needs to be changes as previously recommended. The bias needs to be acknowledged between the open and arthroscopic techniques.

4) The authors state that consent was obtained in answering this reviewer's question but they didi not put this into the paper in the Methods section.

Author Response

1) I have inserted the age of the patients in the methods section and moved to the results section.

2) Line 229: Prosthetic replacement, derotation osteotomy of the humeral neck, and humeral head reconstruction with allograft or autograft transplantations have also been proposed for osteochondral defects larger than 45%.          These surgical procedures are indicated for reverse hill-sachs type lesions with associated significant bone loss.

3) I have added in the discussion section the strength and limitation of the study. I placed the bias between open and arthroscopic technique

4) I included in the methods section that the authors obtained consent from the patients studied.
